# NMR Metabolomic Analysis of Skeletal Muscle, Heart, and Liver of Hatchling Loggerhead Sea Turtles (*Caretta caretta*) Experimentally Exposed to Crude Oil and/or Corexit

**DOI:** 10.3390/metabo9020021

**Published:** 2019-01-26

**Authors:** Stasia A. Bembenek-Bailey, Jennifer N. Niemuth, Patricia D. McClellan-Green, Matthew H. Godfrey, Craig A. Harms, Hanna Gracz, Michael K. Stoskopf

**Affiliations:** 1Department of Clinical Sciences, College of Veterinary Medicine, North Carolina State University, Raleigh, NC 27607, USA; jnniemut@ncsu.edu (J.N.N.); matt.godfrey@ncwildlife.org (M.H.G.); caharms@ncsu.edu (C.A.H.); 2Fisheries, Wildlife, and Conservation Biology, College of Natural Resources, North Carolina State University, Raleigh, NC 27695, USA; 3Environmental Medicine Consortium, North Carolina State University, Raleigh, NC 27607, USA; hanka.gracz@gmail.com; 4Center for Marine Sciences and Technology, North Carolina State University, Morehead City, NC 28557, USA; dmcclel@ncsu.edu; 5Sea Turtle Project, North Carolina Wildlife Resources Commission, Beaufort, NC 28516, USA; 6Nicholas School of the Environment, Duke University Marine Lab, Beaufort, NC 28516, USA; 7Department of Molecular and Structural Biochemistry, North Carolina State University, Raleigh, NC 27607, USA

**Keywords:** *Caretta caretta*, Corexit 9500A, crude oil, heart, liver, metabolomics, NMR, sea turtle, skeletal muscle

## Abstract

We used nuclear magnetic spectroscopy (NMR) to evaluate the metabolic impacts of crude oil, Corexit 5900A, a dispersant, and a crude oil Corexit 5900A mixture exposure on skeletal muscle, heart, and liver physiology of hatchling loggerhead sea turtles (*Caretta caretta*). Tissue samples were obtained from 22 seven-day-old hatchlings after a four day cutaneous exposure to environmentally relevant concentrations of crude oil, Corexit 5900A, a combination of crude oil and Corexit 9500A, or a seawater control. We identified 38 metabolites in the aqueous extracts of the liver, and 30 metabolites in both the skeletal and heart muscle aqueous extracts, including organic acids/osmolytes, energy compounds, amino acids, ketone bodies, nucleosides, and nucleotides. Skeletal muscle lactate, creatines, and taurine concentrations were significantly lower in hatchlings exposed to crude oil than in control hatchlings. Lactate, taurine, and cholines appeared to be the basis of some variation in hatchling heart samples, and liver inosine, uracil, and uridine appeared to be influenced by Corexit and crude oil exposure. Observed decreases in concentrations of lactate and creatines may reflect energy depletion in skeletal muscle of oil-exposed animals, while decreased taurine concentrations in these animals may reflect higher oxidative stress.

## 1. Introduction

Sea turtles are long-lived, large-bodied animals that move through many habitats throughout their lives [1]. They are intrinsically valuable to marine trophic webs and extrinsically valuable to humans (e.g., ecotourism) [2]. All recognized species worldwide have populations that are classified as vulnerable to critically endangered, and most are listed or protected under international conservation treaties and agreements, including the International Union for Conservation of Nature [3]. There is increasing concern that environmental contaminants including crude oil and dispersants (e.g., Corexit 5900A, hereafter referred to as Corexit) could enhance declines of already vulnerable sea turtle populations [4]. While health effects of oil exposure to sea turtles at high doses have been reported (e.g., in Reference [5]), the mechanisms of pathogenesis, and sub-lethal and subclinical effects of exposure are not well understood. 

Metabolomics is a sensitive method to assess sub-lethal and subclinical effects of chemical exposure [6] and provides insight into the health status of an organism [7]. Nuclear magnetic resonance spectroscopy (NMR) has the advantage of being a nondestructive and nonselective modality that is highly quantitative and reproducible [8]. The characterization of tissue metabolic signatures of sea turtles exposed to crude oil and Corexit could enhance understanding of how crude oil and Corexit affect sea turtle physiology. Fish exhibit changes in metabolite concentrations in skeletal muscle and liver after exposure to crude oil and Corexit [9,10]. Environmentally relevant exposures of crude oil have also resulted in alterations in heart morphology and function in embryonic fish [11]. Little is known of the toxic effects of crude oil or Corexit on skeletal muscle, heart, or liver in other taxa [12], including sea turtles. While polycyclic hydrocarbons from oil exposure are absorbed into the liver of sea turtles in the wild [5], there is minimal information of how the crude oil and Corexit exposure affect the liver physiology of sea turtles (e.g., [13,14]).

In this study, we compared metabolic signatures of hatchling loggerhead sea turtle (*Caretta caretta*) skeletal muscle, heart, and liver tissues from animals experimentally exposed to crude oil, Corexit, a combination of crude oil and Corexit, or a seawater control. We analyzed the metabolic signatures of both the aqueous and lipophilic extracts of the tissues, and assessed for differences between exposure groups within a tissue type. Analyzing both aqueous and lipophilic fractions of the tissue extraction allowed for a more complete assessment of any impact of crude oil and/or Corexit exposure on hatchling physiology, and a more comprehensive metabolic signature of the hatchlings. 

## 2. Results

### 2.1. Metabolic Profiling

#### 2.1.1. Aqueous Data

We identified 30, 30, and 38 metabolites from distinct peaks visible in the spectra of hatchling loggerhead sea turtle skeletal muscle, heart, and liver (Figure 1; Table 1), respectively, using 1D ^1^H and 2D (^1^H-^1^H) and (^1^H-^13^C) techniques. These represented 53 unique metabolites. Metabolite classes observed included organic acids/osmolytes, energy compounds, amino acids, a ketone body, and nucleosides and nucleotides. Eighteen metabolites were present in all three tissue types. Eight metabolites were identified only in skeletal muscle, including acetate, carnosine, formate, myo-inositol, maltose, histidine, inosine, and inosine monophosphate. Seven metabolites were identified only in heart, including creatinine, glutathione, glycogen, adenine, adenosine, adenosine diphosphate, and nicotinamide adenine dinucleotide. Fifteen metabolites were identified only in liver, including uridine, uracil, cytidine, 2-hydroxyvalerate, 2-hydroxybutyrate, pyruvate, glycylproline, methylamine, tyrosine, threonine, phenylalanine, alloisoleucine, an alanine dimer, arginine, and mannose. Each metabolite identified occurred in all samples of a given tissue type, regardless of treatment group. 

#### 2.1.2. Lipophilic Data

In the lipophilic fractions of the tissues, we identified triacylglycerides (TAGs), diacylglyderides (DAGs), choline, cholesterol, esterified cholesterol, lathosterol, and saturated and unsaturated fatty acids (FAs), including omega-3 fatty acids (Figure 2). In the lipophilic fraction of skeletal muscle, there were 2 different DAGs, one of which contained an omega-6 FA (linoleic acid), and 3 different unsaturated FAs (Appendix A). In the heart lipophilic fraction, there were at least 3 different unsaturated FAs, including an omega-3 FA, but no linoleic acid (see Appendix A). In the liver lipophilic fraction, unsaturated fatty acids included omega-3 (docosahexaenoic acid, eicosapentaenoic acid, alpha-linoleic acid) and omega-6 fatty acids (linoleic acid) (Appendix A). TAGs were identified in a higher relative concentration to other metabolites (e.g., lathosterol, DAGs, and choline), whereas the choline was present in a lower relative concentration in the liver than in the skeletal muscle and heart (Figure 2).

### 2.2. Statistical Analysis

#### 2.2.1. Aqueous Data

The principal components analysis (PCA) of the aqueous skeletal muscle, heart, and liver normalized and weighted integrals did not show any grouping of samples based on treatment. A single hatchling (B45) exposed to Corexit and crude oil separated from the other samples in the PCA, particularly in the heart and liver samples. This sample had elevated lactate (heart and liver) and lower sugars (liver) in comparison to other samples of the same tissue type (Appendix A). The heat maps suggested a possible trend of skeletal muscle catabolism for most polar compounds detected in samples from hatchlings exposed only to crude oil (Appendix A). This change was not readily apparent in skeletal muscle of hatchlings exposed to both crude oil and Corexit. No similar pattern was clear in the heat map of polar compounds from cardiac muscle (Appendix A). The heat map of polar compounds from hatchling liver was distinguished by apparent increased small molecule anabolism in animals exposed to both crude oil and Corexit, and a decrease in pyrimidine and purine metabolites in animals exposed to crude oil and Corexit separately (Appendix A).

Overall in the polar tissue extracts, there were no fold differences between treatment groups. Only 3 of the identified metabolites varied in a statistically significant manner across groups (α = 0.05). Lactate (combined 1.32–1.37 ppm and 4.09–4.13 ppm), creatines (combined 3.01–3.06 ppm and 3.91–3.95 ppm), and taurine (combined 3.24–3.29 ppm and 3.41–3.46 ppm) integrals were significantly different among treatment groups in skeletal muscle (Kruskal Wallis tests [KW]; *p* = 0.020, *p* = 0.016, *p* = 0.028, respectively; Figure 3). Post hoc testing (Dunn’s all-pairwise comparisons tests) revealed that the concentrations of all three skeletal muscle metabolites were significantly lower in hatchlings exposed to crude oil only, compared to control hatchlings. There were no statistically significant differences among the treatment groups for any of the heart metabolites, but Corexit exposure appeared to promote lower concentrations of lactate, cholines, and taurine, and narrower concentration ranges of these metabolites (Appendix A). Some differences between treatment groups, albeit not statistically significant, were identified in the liver for nucleobase uracil and the nucleosides inosine and uridine (Appendix A) (KW; *p* = 0.073, *p* = 0.080, *p* = 0.068). Exposure to crude oil, and in particular, Corexit, appeared to be associated with lower concentration of pyrimidine and purine metabolites, as well as reduced variation between samples. 

#### 2.2.2. Lipophilic Data

The PCA of the lipophilic skeletal muscle, heart, and liver normalized and weighted integrals did not show any grouping of samples based on treatment. Heat maps of the functional group integrals from identified compounds illustrated variation across samples (Appendix A), but clear patterns across treatment groups were not readily apparent. Relative amounts of TAGs and FAs were determined by comparing peak-specific integral ratios between groups. No statistically significant differences (α = 0.05) were detected between treatment groups of the skeletal muscle, heart, and liver when comparing ratios of integrals from TAGs to total terminal methyl groups, nor the ratios of integrals from unsaturated FAs to those from the terminal methyl groups from any FAs. 

## 3. Discussion

The use of 2D methods along with the 1D ^1^H NMR spectra allowed identification of 53 metabolites in the aqueous fractions, with equal number of metabolites identified in the skeletal muscle and heart (n = 30), and a higher number in the liver (n = 38). As expected, considering the comparable contractile apparatus in the muscle types [15], the metabolomes of the skeletal muscle and heart were similar. Differences were noted in both nucleotide metabolites and in the amino acid profiles. Adenine nucleotide degradation in mammals occurs mainly through the inosine monophosphate pathway in skeletal muscle, and through conversion to adenosine and adenine in the heart [16]. Similar purine degradation pathways appear to be present in the hatchling turtles. 

The heat map data for polar compounds showed us metabolic depletions in the skeletal muscle of hatchlings exposed to crude oil. These hatchlings had significantly lower skeletal muscle concentrations of lactate, creatines, and taurine than control animals. Lactate [17] and creatines [18] can be energy sources for muscle, and a lower concentration may reflect energy depletion in hatchlings exposed to crude oil. Lactate is also a product of glycolysis under anaerobic conditions [19]. With an increased energy demand, lactate may have been shuttled to the citric acid cycle [17] and to glycogen replacement in the muscle [20]. It is also possible that there was a toxic effect of the crude oil on the function of lactate dehydrogenase (catalyzes the reduction of pyruvate to lactate), which may also have decreased the lactate in the skeletal muscle. In Atlantic salmon (*Salmo salar*), crude oil exposure caused an inhibition of gill lactate dehydrogenase [21]. These factors alone or in combination could have led to a lower lactate concentration in the skeletal muscle of crude-oil-exposed hatchlings. 

Creatines are essential for maintenance of an energy buffer in tissues [22]. A lower concentration of creatines may reflect energy depletion in the hatchling skeletal muscle. This may have occurred because of an increased conversion to creatinine that was eliminated from the muscle [18]. It is also possible that the crude oil exposure affected the production of creatine in the liver and kidney, decreasing the amount available for skeletal muscle [18].

Taurine is an essential amino acid that has an osmoregulatory role in cells and is reported to have antioxidant effects [23]. Lower muscle concentrations may reflect osmotic and oxidative stresses caused by crude oil exposure. A significantly lower taurine concentration in the skeletal muscle may have also occurred because of an enzymatic inhibition of taurine production [24]. Increased utilization of hypotaurine, the immediate precursor of taurine, as an antioxidant in the face of crude oil exposure is another possibility [24]. 

Similar effects of oil exposure were not seen in heart samples. There appeared to be some effect of Corexit exposure that resulted in lower concentrations of lactate, taurine, and cholines (Appendix A). Any toxic effect of Corexit was not seen statistically, potentially in part due to the small sample size of the treatment groups and individual variation. However, due to the limited body of work on effects of Corexit exposure alone in wildlife, the findings represent a starting point for future work. 

Crude oil and Corexit appeared to impact purine and pyrimidine metabolism in the liver. Lower concentrations of liver inosine, uracil and uridine were observed in hatchlings exposed to crude oil and Corexit. Additionally, the range of these metabolite concentrations narrowed with Corexit and crude oil exposure (Appendix A). Inosine, a purine nucleoside, may have been utilized to a greater degree in the purine salvage reaction to increase production of ATP [25]. As adenosine is a precursor to inosine, it may also be that there was less adenosine available, either from loss through extracellular transport or decreased degradation from AMP. More definitive determination of the cause of a lower inosine concentration is problematic, as other metabolites in the purine salvage reaction were not identified in the liver spectra. While ATP was identified in the liver ^1^H NMR spectra, it was as at low concentrations across all treatment groups, such that meaningful evaluation of any impact was not possible. A demand for ATP may also have led to an increase in uridine and uracil catabolism to feed the citric acid cycle with more acetyl CoA. Also possible is their conversion to RNA or DNA via thymidine 5’P. The purpose of this mechanism is not clear, except to perhaps replace live DNA/RNA damaged by crude oil and Corexit exposure. 

Acute death was rare in our study. The smallest hatchling (B45) in the study died during the final processing. That hatchling had been exposed to both Corexit and crude oil, and had a substantially higher lactate in the heart than other hatchlings (Appendix A). It is possible that the crude oil and Corexit treatment induced a cardiotoxicity similar to those demonstrated in embryonic fish [26,27]. 

The lipophilic fraction is frequently lost due to processing technique or not described in ^1^H NMR studies of animals. While it is challenging to identify specific metabolites in lipophilic fractions using ^1^H NMR, our study showed that there are interesting similarities and differences in the lipophilic fractions that can further inform us about the metabolism of the tissue types. Specifically, TAGs were identified in a higher relative concentration to other metabolites in the liver than in the skeletal muscle and heart. This difference may be because the liver is the major site of TAG synthesis in reptiles [28,29]. However, the precise role of the liver in hatchling TAG production and storage is not known.

Many metabolites, including all of the nucleotides and nucleosides, were present in low concentrations in the tissues (Figure 1, Figure 2 and Figure 3). Additional metabolites may have been identified with larger tissue samples, as we observed with the liver, which may have better informed differences across exposure groups. After analyzing skeletal muscle control samples individually, we pooled remaining unprocessed skeletal muscle from multiple animals to increase the concentration of metabolites; this allowed us to perform additional 1D and 2D NMR experiments for metabolite identification and confirmation. However, the hearts were so small that the entire organ was used in the original processing, precluding the opportunity to directly increase metabolite concentrations through pooling of minimally processed samples. 

Although the hatchlings in our study were exposed to environmentally relevant concentrations of crude oil and Corexit designed to replicate a spill and attempted remediation, the lack of the environmental influences on exposures may have decreased the potential for additional observed toxic effects. The loggerhead hatchlings in our study were held under controlled conditions eliminating many environmental variables (e.g., wave action, ambient temperatures of water and air, oxygen level) that could have influenced bioavailability of the toxic elements in the crude oil and Corexit [30]. In addition to bioavailability, environmental circumstances could result in a threshold effect that was not replicated in our controlled laboratory study that resulted in evident toxicological effects in studies involving other species (e.g., cold temperatures [31]). Furthermore, the duration of exposure was limited, and a longer exposure may have resulted in greater effects.

This NMR study described the hatchling loggerhead skeletal muscle, heart, and liver metabolic signatures, identified physiological effects of crude oil exposure on hatchling sea turtle skeletal muscle metabolism, and suggested impacts of crude oil and Corexit exposure on liver metabolism. Observed decreases in concentrations of lactate and creatines may reflect energy depletion in skeletal muscles of exposed animals, while decreased taurine concentrations in these animals may reflect higher oxidative stress. A few interesting metabolite patterns were suggestive of a potential cardiotoxicity of Corexit, and impacts on liver purine and pyrimidine metabolism in Corexit and crude oil exposed animals, but no significant differences were found in either tissue from animals exposed to Corexit, crude oil, or a combination of crude oil and Corexit when compared to unexposed animals. This work demonstrated that valuable information can be gleaned from small tissue samples using NMR metabolomic methods, and represents a foundation for future toxicological and physiological studies of loggerhead sea turtles and other endangered wildlife.

## 4. Materials and Methods

### 4.1. Organisms, Oil/Dispersant Exposure, and Tissue Collection

Loggerhead sea turtle eggs (n = 22) were collected from nests certain to fail due to timing of lay, nest position, expected weather impacts, and predation from invasive species. Using eggs from a single nest provided the advantage of minimal genetic variation amongst the animals in much the way that inbred strains of laboratory animals are valuable in medical research. Experimental use of the animals was approved by Endangered Species Permit 13ST50 from the North Carolina Wildlife Resources Commission with federal authority delegated from the United States Fish and Wildlife Service and the North Carolina State University Institutional Care and Use Committee (11-078-O, 11-103-O). In the laboratory, the eggs were incubated under controlled conditions of 27.2–30.8 °C. Three days post-hatch, the turtles were exposed to crude oil, Corexit, a combination of crude oil and Corexit, or a negative control for 4 days, as described in our previous study [32]. Following euthanasia of each hatchling, biofluids and tissues, including liver, and pectoralis (skeletal) and heart muscles, were excised, flash frozen directly in wells of dry ice, placed into labeled polyethylene freezer tubes (Thermo Scientific Nalgene, Waltham, MA, USA), and stored at −80 ^°^C. A scheme illustrating the overall design of our study is presented in Appendix A.

### 4.2. Metabolite Extraction

Frozen skeletal muscle, heart, and liver samples were weighed (average wet tissue weights (mg) ± standard deviation were 49.6 ± 8.9, 135.2 ± 57.6, and 183.8 ± 39.5, respectively) and homogenized by adding a 2:1 (*v*/*w*; 2 mL solution to 1 g tissue) amphibian Ringer’s solution (Fisher Scientific, Waltham, MA, USA) and ~1:1 (*v*/*v*) 0.5 mm zirconium oxide beads to the tissue in microtubes (Xtreme Series Ultra-Clear Microtubes, Phenix Research Products, Candler, NC, USA) and processed on speed eight for 3 min in a homogenizer (Bullet Blender, Next Advance, Averill Park, NY, USA). The homogenized samples were centrifuged at 12,848× *g* for 25 min and the supernatant was pipetted into a new polypropylene tube. The pellets and supernatants were stored at −80 °C until further processing.

To better capture both aqueous and non-aqueous metabolites in the heart and skeletal muscle, we recombined the pellet and supernatant by adding 0.5 mL ultrapure water to the pellet, vortexing, and adding the pellet and its supernatant to the previous supernatant. This was repeated four times (until the rinse appeared clear and colorless) for each sample. The recombined homogenates were lyophilized overnight (Labconco FreeZone 2.5 Liter Freeze Dry System, Series 74200, −81.3 °C). 

Approximately half of the lyophilized skeletal muscle sample was returned to the −80 °C freezer, while the other half was weighed and placed in a borosilicate glass centrifuge tube (Fisherbrand, Waltham, MA, USA) and processed further. The entire heart was processed to maximize data acquisition. Chilled methanol (24 mL/g lyophilized tissue) and ultrapure water (9 mL/g lyophilized tissue) were added to each sample, and the mixture was vortexed for 15 s. Chilled chloroform (48 mL/g lyophilized tissue) and ultrapure water (9 mL/g lyophilized tissue) were then added, followed by a 60 s vortex and 10 min on wet ice. The mixtures were centrifuged at 1690× *g* for 10 min. The aqueous layers were pipetted into polypropylene tubes, frozen on dry ice, lyophilized, and stored at −80 °C until NMR analysis. The lipophilic layers from the skeletal muscle samples were pipetted into 16 × 125 mm borosilicate glass tubes (Fisherbrand, Waltham, MA, USA), covered with aluminum foil and Parafilm (Pechiney Plastic Packaging, Chicago, IL, USA) and temporarily stored at −80 °C. The samples were subsequently dried under a stream of nitrogen at 30 °C (Labconco Rapidvap Vertex Evaporator, Kansas City, MO, USA), covered with aluminum foil and Parafilm and stored at −20 °C. The lipophilic layers from the heart samples were similarly processed, dried and stored without the step of being stored at −80 °C between processing and drying. 

The liver pellet and supernatant were similarly recombined, and the tube that contained the pellet was rinsed four times with ¼ of the chilled methanol required for further processing (1.33 mL/1 mL of amphibian Ringer’s solution used in the original processing) and vortexed for 15 s. Chilled chloroform (5.33 mL/1 mL of amphibian Ringer’s solution) and ultrapure water (1 mL/1 mL of amphibian Ringer’s solution) were then added, followed by a 60 s vortex and 10 min on wet ice. The mixtures were centrifuged at 1690 g for 10 min. The aqueous and lipophilic layers were pipetted into separate 16 × 125 mm borosilicate glass tubes, and dried and stored as the lipophilic heart and skeletal muscle extracts. 

### 4.3. ^1^H-NMR Metabolomics and Spectral Pre-Processing

Immediately prior to NMR analysis, the sealed aqueous extracts of skeletal muscle were thawed at room temperature, dissolved in 35 μL 100% D_2_O containing 0.1 mM trimethylsilyl propionate (TSP), 1 mM formate, 40 mM phosphate buffer, and 0.05% sodium azide, and filtered (Fisherbrand SureOne 10 μL, extended, filter, low retention, universal fit pipet tips, Thermo Fisher Scientific, Waltham, MA, USA) at 3000 g for 2 min. One-dimensional ^1^H-NMR spectra of the extracts were obtained using a microcoil NMR probe (Protasis, Marlboro, MA, USA) in a Varian Inova 600 MHz multinuclear INOVA NMR spectrometer (Varian, Inc., Palo Alto, CA, USA) at 25 °C with a 1.1 s acquisition time. The sweep width of 7193.6 Hz acquired 8192 complex points and 5120 transients. 

The sealed aqueous heart and liver extracts were dissolved in 250 μL and 300 μL 10% D_2_O/90% H_2_O, respectively, containing 0.1 mM TSP, 40 mM phosphate buffer, and 0.05% sodium azide, and pipetted in 3 mm NMR tubes (327PP-7; Wilmad-LabGlass, Vineland, NJ, USA). The spectra were obtained using a 5 mm ID ^1^H/BB (^109^Ag-^31^P) Triple-Axis Gradient Probe (ID500-5EB, Nalorac Cryogenic Corp.) in a Bruker 500 MHz NMR spectrometer (Bruker Biospin, Billerica, MA, USA) with a 2.0447 s acquisition time. The sweep width of 8012.58 Hz acquired 32,768 complex points and 256 transients.

The lipophilic extracts were thawed at room temperature and dissolved in 250 μL (skeletal muscle and heart) or 300 μL (liver) of deuterated chloroform +/− 0.03% tetramethylsilane (TMS) (Sigma-Aldrich Co., St. Louis, MO, USA), which was used as an internal standard. The spectra were obtained using the same spectrometer as the aqueous heart and liver extracts, with some differences in acquisition time (1.6384 s for skeletal muscle), sweep width (9999.39 Hz for skeletal muscle), complex points (16,384 for skeletal muscle and liver), and number of transients (128 for heart). 

### 4.4. 2D NMR and 1D ^31^P NMR Methods

To aid identification and confirmation of metabolites, 2D (^1^H-^1^H), (^1^H-^13^C) and 1D ^31^P experiments were performed and analyzed on the aqueous and lipophilic liver, skeletal, and heart muscle extracts. As individual skeletal muscle samples from the hatchling loggerheads were too small for adequate 2D and 1D ^31^P NMR analysis, the remaining portion of the 6 homogenized and lyophilized skeletal muscle samples from the control hatchling loggerheads were combined and processed as heart samples. The experimental details of the 2D (^1^H-^1^H) correlation spectroscopy (COSY), (^1^H-^1^H) total correlation spectroscopy (TOCSY), (^1^H-^13^C) heteronuclear single quantum correlation (HSQC), (^1^H-^13^C) heteronuclear multiple bond correlation (HMBC), 1D ^31^P, and repeat 1D ^1^H NMR experiments are presented in Appendix B. 

### 4.5. NMR Data Processing and Analysis

The number of samples included in the NMR data analysis were: 22 aqueous and 21 lipophilic skeletal muscle extracts, 21 aqueous and 22 lipophilic heart extracts, and 21 aqueous and lipophilic liver extracts. Sample extracts that were lost during NMR analysis (n = 1) or had poor peak definition and shape relative to other spectra (n = 3) were not included in the data analysis. The aqueous data were pre-processed using ACD Labs 12.0 1D NMR Processor (Advanced Chemistry Development, Toronto, ON, Canada). This included zero-filling to 16,384 points, phase correction, baseline correction, and alignment of the reference TSP signal in the samples. For data analysis, dark regions were created where there was residual water and an absence of peaks of interest. The data were integrated using intelligent bucket integration with a bin width of 0.03 ppm with 50% looseness. For the liver samples, we manually integrated the peaks to reduce the impact of non-uniform peak shifts. The integrals for each sample were normalized to the TSP integral and weighted by the sample mass. There were 91 bins for skeletal muscle, 81 bins for heart, and 27 bins for liver aqueous extracts. Peak identification was performed using Chenomx NMR Suite 8.1 (Chenomx, Edmonton, AB, Canada), the Human Metabolome Database (HMDB; [33]), as well as 2D NMR experimental data, 1D ^31^P NMR experiments and NMR standards created in our laboratory. Where metabolite peaks overlapped and the peak integrals could not be precisely differentiated, we grouped similar metabolites together (e.g., cholines including choline, phosphocholine, and glycerophosphocholine, creatines including creatine and phosphocreatine). Multiple, nonoverlapping integrals belonging to particular metabolites (e.g., lactate) were combined to form a metabolite integral proportional to metabolite concentration. 

The lipophilic data were processed similarly, with the exception that alignment was to the chloroform signal. The manually integrated peaks for each sample were normalized by dividing each integral by the sum of the integrals of a given bin. There were 23 bins for skeletal muscle, 26 bins for heart, and 29 bins for the liver lipophilic extracts. Signal identification was performed using HMDB, standards created in our laboratory, predicted spectra (ACD SACD11/C+H Predictor and DB), the American Oil Chemists’ Society (AOCS) Lipid Library [34], 2D data and available literature [35,36,37,38].

### 4.6. Statistical Analysis of NMR Data

PCA was performed using normalized and weighted integrals from aqueous extracts of the tissue samples with JMP Pro12 (SAS Institute, Inc., Cary, NC, USA). Heat maps were generated using normalized and weighted integrals from the aqueous extracts with JMP Pro12 (SAS Institute, Inc., Cary, NC, USA). Differences in metabolite integrals across treatment groups were assessed using Kruskal Wallis tests (STATISTIX 10, Statistix, Inc., Tallahassee, FL, USA; α = 0.05). Dunn’s all-pairwise comparisons test (STATISTIX 10, Statistix, Inc., Tallahassee, FL, USA; α = 0.05) was used as a post hoc assessment for differences between treatment groups. As comparing statistical significance using *p* values can have limited value with small treatment group sizes [39], we examined the distribution and variability of the identified metabolite peak integrals in individual value plots to assess for biological effects of exposure. 

PCA was performed and heat maps were generated using the normalized and weighted integral data from the lipophilic tissue extracts as with the aqueous extracts. Additionally with the lipophilic integral data, we calculated the ratios of integrals of protons in β-position in relation to alkene (1.92–2.15 ppm) to the integrals of terminal –CH_3_ protons (0.75–1.05 ppm), to gauge any decrease in unsaturated FAs in comparison to the total FAs present in the sample. To evaluate a potential decrease in triacylglycerides (TAGs) as they break down in response to increased energy demands, we assessed the ratio of TAGs to the total terminal –CH_3_ by comparing ratios of integrals of protons of the alpha glycerol backbone of TAGs (4.1–4.4 ppm) to the integrals of terminal –CH_3_ protons. Differences in ratios across treatment groups were assessed using Kruskal Wallis tests (STATISTIX 10, Statistix, Inc., Tallahassee, FL, USA; α = 0.05). 

## Figures and Tables

**Figure 1 metabolites-09-00021-f001:**
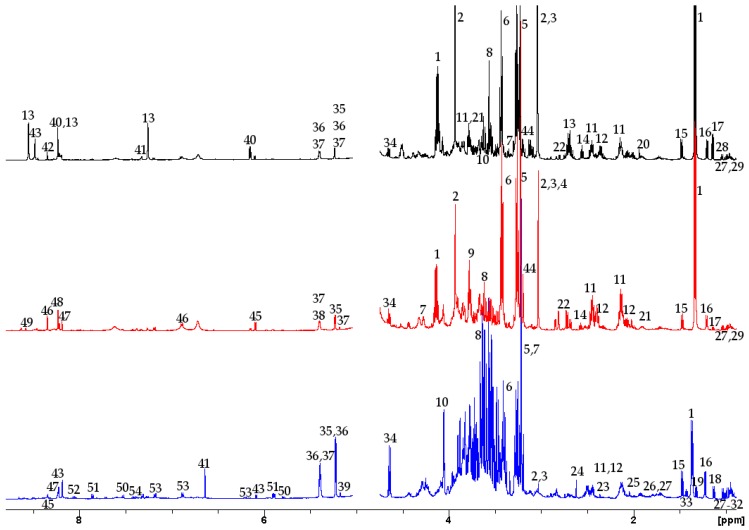
Representative ^1^H-NMR spectra of aqueous extracts of skeletal muscle (top/black), heart (middle/red), and liver (bottom/blue) from hatchling loggerhead sea turtles (*Caretta caretta*). The y axis is indicative of peak intensity, which correlates to concentration. Peaks 1, 2, 3, and 5 are truncated. The water peak has been removed. Labeled peaks are as follows: (1) lactate, (2) creatine, (3) phosphocreatine, (4) creatinine, (5) phosphocholine, (6) taurine, (7) glycerophosphocholine, (8) glycine, (9) glutathione, (10) myo-inositol, (11) glutamine, (12) glutamate, (13) carnosine, (14) beta-alanine, (15) alanine, (16) 3-hydroxybutyrate, (17) propylene glycol, (18) unknown, (19) threonine, (20) acetate, (21) homoserine, (22) aspartate, (23) pyruvate, (24) methylamine, (25) glycylproline, (26) arginine, (27) leucine, (28) valine, (29) isoleucine, (30) alloisoleucine, (31) 2-hydroxybutyrate, (32) 2-hydroxyvalerate, (33) alanine dimer, (34) beta-glucose, (35) alpha-glucose, (36) maltose, (37) ribose, (38) glycogen, (39) mannose, (40) inosine monophosphate, (41) histidine, (42) formate, (43) inosine, (44) choline, (45) adenosine triphosphate, (46) nicotinamide adenine dinucleotide, (47) adenine, (48) adenosine, (49) adenosine diphosphate, (50) uracil, (51) uridine, (52) cytidine, (53) tyrosine, (54) phenylalanine. The uncropped, labeled spectra are in Appendix A.

**Figure 2 metabolites-09-00021-f002:**
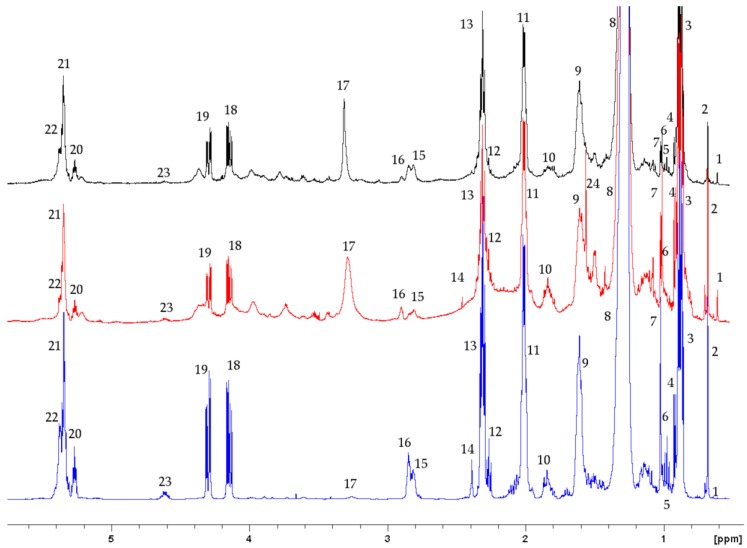
Representative ^1^H-NMR spectra of lipophilic extracts of skeletal muscle (top/black), heart (middle/red), and liver (bottom/blue) from hatchling loggerhead sea turtles (*Caretta caretta*). The y axis is indicative of peak intensity, which correlates to concentration. Peaks 3 and 8 are truncated. Key (the relevant protons are bolded): (1) lathosterol –C**H_3_**, (2) total cholesterol –C**H_3_**, (3) fatty acids (except omega-3) –C**H_3,_** (4) total cholesterol –C**H_3_**, (5) fatty acids (omega-3) –C**H_3_**, (6) free cholesterol –C**H_3_**, (7) esterified cholesterol –C**H_3,_** (8) fatty acids (except 20:5 omega-3 and 22:6 omega-3) –(C**H_2_**)_n_–, (9) fatty acids (except 20:5 omega-3 and 22:6 omega-3) –C**H_2_**–CH_2_–COOH, (10) fatty acids (20:5 omega-3 and 22:6 omega-3) –C**H_2_**–CH_2_–COOH, (11) unsaturated fatty acids –C**H_2_**–CH=CH, (12) acyl group in triacylglycerides –C**H_2_**–COOH, (13) acyl group in fatty acids (except 22:6 omega-3), monoacylglycerides and diacylglycerides –RH–C**H_2_**–COOH, (14) fatty acid (22:6 omega-3) –C**H_2_**–COOH (15) fatty acid (18:3 omega-3) =CH–C**H_2_**–CH=, (16) polyunsaturated fatty acids =CH–C**H_2_**–CH=, (17) choline –N(C**H_3_**)_3_, (18) glyceryl protons of triacylglycerides and diacylglycerides ROC**H_2_**–CH(OR’) –C**H_2_**OR”/ROC**H_2_**–C**H**OH–C**H_2_**OR’, (19) glyceryl protons of triacylglycerides and diacylglycerides ROC**H_2_**–CH(OR’)–C**H_2_**OR”/ROC**H_2_**–C**H**OH–C**H_2_**OR’, (20) glyceryl protons in triacylglycerides ROCH_2_–C**H**(OR’)–CH_2_OR”, (21) unsaturated fatty acids –C**H**=C**H**– *cis*, (22) unsaturated fatty acids –C**H**=C**H**– *trans*, (23) cholesterol ester –C**H**–, and (24) water. Metabolite chemical shifts and multiplicities are provided in Appendix A.

**Figure 3 metabolites-09-00021-f003:**
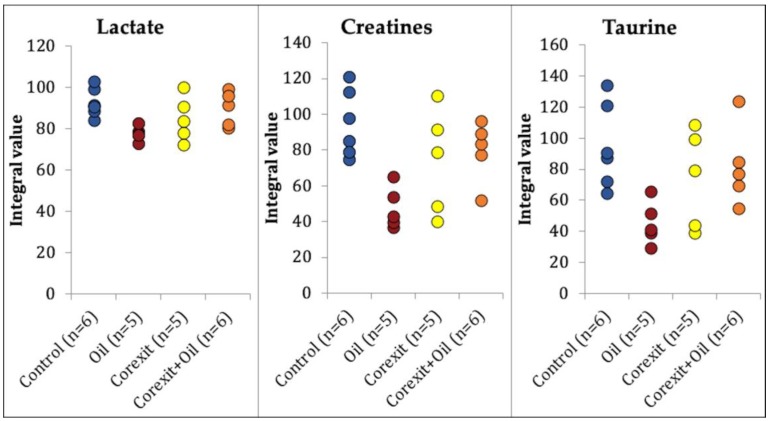
Individual value plots of the normalized and weighted integrals of lactate, creatines, and taurine from the aqueous extracts of hatchling loggerhead sea turtle (*Caretta caretta*) skeletal muscle. Integral values are analogous to metabolite concentration. Treatment groups are labeled on the x axis. Lactate, creatines, and taurine concentrations were significantly different among treatment groups (Kruskal Wallis tests: *p* = 0.020, *p* = 0.016, *p* = 0.028, respectively, α = 0.05). Post hoc Dunn’s all-pairwise comparisons tests showed that control and crude oil groups were significantly different from each other in lactate, creatines, and taurine.

**Table 1 metabolites-09-00021-t001:** Metabolites identified in the aqueous extracts of skeletal muscle, heart, and liver of hatchling loggerhead sea turtles (*Caretta caretta*) grouped by metabolite type. Key: Skeletal muscle (S), Heart (H), Liver (L); (*) Confirmed using 2D NMR +/− 1D ^31^P NMR experiments, laboratory standards +/− predicted spectra (ACD SACD11/C+H Predictor and DB). (#) Metabolites that met the statistical significance threshold (α = 0.05). Metabolite chemical shifts and multiplicities are provided in Appendix A.

Metabolite	Tissue	Metabolite	Tissue
***Organic acids/Osmolytes***
Acetate	S*	Glutathione	H
Carnosine	S	Glycerophosphocholine	S, H, L*
Choline	S, H, L*	Lactate	S^#^, H, L*
Creatinine	H*	Myo-inositol	S, L*
Formate	S	Phosphocholine	S, H, L*
***Energy compounds***
Creatine	S^#^, H, L*	Mannose	L
Glucose (α & β)	S, H, L*	Ribose	S, H, L*
Glycogen	H*	Phosphocreatine	S^#^, H, L*
Maltose	S, L*		
***Amino acids***
Alanine	S, H, L*	Histidine	S, L *
Alanine dimer	L*	Homoserine	S, H
Alloisoleucine	L*	Isoleucine	S, H, L*
Arginine	L*	Leucine	S, H, L*
Aspartate	S, H*	Methylamine	L*
Beta-alanine	S, H*	Phenylalanine	L*
Glutamate	S, H, L*	Taurine	S^#^, H, L*
Glutamine	S, H, L*	Threonine	L*
Glycine	S, H, L*	Tyrosine	L*
Glycylproline	L*	Valine	S, H, L*
***Ketone bodies***
3-hydroxybutyrate	S, H, L*	2-hydroxyvalerate	L
2-hydroxybutyrate	L*	Pyruvate	L*
***Nucleosides & Nucleotides***
Adenine	H, (L*)	Inosine	S, L*
Adenosine	H	Inosine monophosphate	S
Adenosine diphosphate	H	Nicotinamide adenine dinucleotide	H
Adenosine triphosphate	H, L*	Uracil	L*
Cytidine	L*	Uridine	L*
***Other***
Propylene glycol	S, H*

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
