# Peer review of "NMR Metabolomic Analysis of Skeletal Muscle, Heart, and Liver of Hatchling Loggerhead Sea Turtles (Caretta caretta) Experimentally Exposed to Crude Oil and/or Corexit"

_metabolites, 2019, doi:10.3390/metabo9020021_

Round 1
Reviewer 1 Report
The authors have determined the metabolic effects of crude oil and/or Corexit on sea turtles by a comprehensive NMR profiling of tissue-extracts. The significant results are shown in Fig 6, indicating a lower level of creatine, lactate and taurine in the exposed sea turtles.
Overall, the manuscript is well-written with clear and concise methodologies and discussions of the approach. The study provides a sound basis of the NMR fingerprints for studying sea turtles.
One comment: HR-MAS could be of interest in the study. This would prevent from the loss of the info on the lipophilic factions (i.e. no sample-processing is necessary). The 1H-profile from the extracts could be an aid in the analysis of HR-MAS spectra; moreover, it only requires only a few mg of tissue (a much smaller mass as compared to that of the tissue requires for processing the extracts).
Author Response
Point 1:HR-MAS could be of interest in the study. This would prevent from the loss of the info on the lipophilic factions (i.e. no sample-processing is necessary). The 1H-profile from the extracts could be an aid in the analysis of HR-MAS spectra; moreover, it only requires only a few mg of tissue (a much smaller mass as compared to that of the tissue requires for processing the extracts).
Response 1:Thank you for your positive review of our paper. We agree that HR-MAS would provide the insights you suggest but unfortunately, we do not have access to an HR-MAS probe. In addition, our very small samples from these hatchings are insufficient for further experiments. We appreciate the thought but are unable to carry out those experiments.
Reviewer 2 Report
The manuscript “NMR metabolomic analysis of skeletal muscle, heart and liver of hatchling loggerhead sea turtles (Caretta caretta) experimentally exposed to crude oil and/or Corexit” by Bembenek‑Bailey and colleagues investigates, by NMR metabolomics on aqueous and lipid extracts, the impact of crude oil and Corexit on skeletal muscle, heart and liver of sea turtles.
The article is well-written and it can be of interest to the readers of Metabolites. However, in my opinion some issues should be addressed before publication.
The first one concerns a couple of things to improve the readability of the manuscript. Due to the many variables present in the study, a scheme illustrating the overall design should be included to allow a better and more immediate understanding of the project.
Also, inclusion of the figures 1-3 in a single one in the main text (same as in Fig. 4 for the lipophilic fractions, should be Fig. S15) would improve the presentation of the article that otherwise might appear too long and difficult to follow because of too many jumps between one paragraph and another. The individual high res spectra, that btw are very nice and useful, should be included in the SI.
The second one regards the statistical analysis of the data. More details about the post-processing of the spectra before the statistical analysis must be included (section 4.6), i.e. manual integration (for what I understood) or binning (what ppm range?), transformation (Pareto, log, mean centering, etc), normalization (total sum?), the dimension of the final aqueous and lipophilic data matrices, etc. Few details are present in the caption of Fig. 5 but should not be there and are not enough.
A table with the average fold-change (or Log2(FC)) for the aqueous and lipophilic metabolites in the 3 groups (O/C/O+C) relatively to the control group should be included in the main text, as well as the corresponding p-values (also if not statistically significant). This data should be embedded in Table 1 if it does not result a too crowded afterwards.
The multivariate analysis lacks many details (as already stated). Furthermore, in the caption of Fig. 5 the PCA plots are reported as score plot (left) and loading plot (right). However, score plots show the samples while loading plots the contribution of the variables to the separation, but in these graphs it is the other way around. This confusion (scores/loadings) is also present in other parts of the text.
The score and loading plots of the corresponding lipophilic extracts should be added to the SI although no separation was observed. Did the univariate analysis confirmed this behavior?
Finally, a heatmap analysis would visually show the variation of the metabolites in the different groups and could improve to understand the behavior of the variables.
Author Response
Point 1:The first one concerns a couple of things to improve the readability of the manuscript. Due to the many variables present in the study, a scheme illustrating the overall design should be included to allow a better and more immediate understanding of the project.
Response 1: Thank you. We created a scheme as suggested and included it as Figure S1 in the Supplementary Materials.
Point 2:Also, inclusion of the figures 1-3 in a single one in the main text (same as in Fig. 4 for the lipophilic fractions, should be Fig. S15) would improve the presentation of the article that otherwise might appear too long and difficult to follow because of too many jumps between one paragraph and another. The individual high res spectra, that btw are very nice and useful, should be included in the SI.
Response 2: Thank you for your suggestion. We replaced Figures 1-3 with a single annotated figure in the main paper (Figure 1) as suggested and placed the individual annotated spectra in Supplementary Materials (Figures S2-S4) to allow readers to view the uncropped spectra.
Point 3: The second one regards the statistical analysis of the data. More details about the post-processing of the spectra before the statistical analysis must be included (section 4.6), i.e. manual integration (for what I understood) or binning (what ppm range?), transformation (Pareto, log, mean centering, etc), normalization (total sum?), the dimension of the final aqueous and lipophilic data matrices, etc. Few details are present in the caption of Fig. 5 but should not be there and are not enough.
Response 3: We included the requested details about post-processing in section 4.5. We felt that position in the paper best fit the flow of the narrative. Integration involved intelligent binning and manual integration along with normalization to TSP (aqueous data) and dividing each integral by the sum of the integrals (lipophilic data). With careful consideration of the many options, analysis without transformation was our preference, which allowed meaningful data interpretation and avoided potential unintended distortions. We added the dimensions of the aqueous and lipophilic matrices as recommended.
The point regarding Figure 5 was inspiring. After looking at recent publications in Metabolitesand long thought and debate among the co-authors, we decided that the PCA analyses did not contribute sufficiently to warrant a figure. We removed Figure 5 and went with simple text as seen in current articles in Metabolites.
Point 4:A table with the average fold-change (or Log2(FC)) for the aqueous and lipophilic metabolites in the 3 groups (O/C/O+C) relatively to the control group should be included in the main text, as well as the corresponding p-values (also if not statistically significant). This data should be embedded in Table 1 if it does not result a too crowded afterwards.
Response 4: We appreciate the suggestion and the need for clarification. We adjusted the text in the manuscript to avoid any confusion that there were fold changes for any examined metabolites (see line 308). There were no fold changes. Table 1 was already quite information dense, so we chose to only mark statistically significantly varying metabolites (p<0.05) in the table and explain this in the table legend.
Point 5:The multivariate analysis lacks many details (as already stated). Furthermore, in the caption of Fig. 5 the PCA plots are reported as score plot (left) and loading plot (right). However, score plots show the samples while loading plots the contribution of the variables to the separation, but in these graphs it is the other way around. This confusion (scores/loadings) is also present in other parts of the text.
Response 5: Please see Response 2 where we addressed the PCA analyses. We decided the PCA analyses did not contribute sufficient information to warrant a figure and removed Figure 5 from the manuscript. We adjusted the language in the manuscript to reflect this (Sections 2.2.1. and 2.2.2.).
Point 6:The score and loading plots of the corresponding lipophilic extracts should be added to the SI although no separation was observed. Did the univariate analysis confirmed this behavior?
Response 6: Please see Responses 2 and 5. Although we did the PCA analyses on the lipophilic extracts, the PCA plots did not substantially contribute to our understanding of the data. After much discussion among co-authors and assessment of current practices in Metabolites, we decided to describe the PCAs in the manuscript text. We did not see any differences between treatment groups in our univariate analyses, and we added text to Section 2.2.2. to make this more clear.
Point 7:Finally, a heatmap analysis would visually show the variation of the metabolites in the different groups and could improve to understand the behavior of the variables.
Response 7: Thank you for the helpful suggestion. We included the heat maps in Figures S18-23 in Supplementary Materials. We added how these were created to our methods (Section 4.6.: lines 696-698 and lines 706-707) and commented on them in the results section (Section 2.2.1.: lines 299-307 and Section 2.2.2.: lines 462-464).
Round 2
Reviewer 2 Report
The revised version of the manuscript is significantly improved and in my opinion it now deserves publication in Metabolites.